# *UBB* pseudogene 4 encodes functional ubiquitin variants

Marie-Line Dubois[1], Anna Meller[1], Sondos Samandi[2], Mylène Brunelle[2], Julie Frion[1], Marie A. Brunet[2], Amanda Toupin[2], Maxime C. Beaudoin[2], Jean-François Jacques [2], Dominique Lévesque[1], Michelle S. Scott[2], Pierre Lavigne[2], Xavier Roucou [2✉] & François-Michel Boisvert [1✉]

Pseudogenes are mutated copies of protein-coding genes that cannot be translated into proteins, but a small subset of pseudogenes has been detected at the protein level. Although ubiquitin pseudogenes represent one of the most abundant pseudogene families in many organisms, little is known about their expression and signaling potential. By re-analyzing public RNA-sequencing and proteomics datasets, we here provide evidence for the expression of several ubiquitin pseudogenes including *UBB* pseudogene 4 (*UBBP4*), which encodes Ub$^{KEKS}$ (Q2K, K33E, Q49K, N60S). The functional consequences of Ub$^{KEKS}$ conjugation appear to differ from canonical ubiquitylation. Quantitative proteomics shows that Ub$^{KEKS}$ modifies specific proteins including lamins. Knockout of *UBBP4* results in slower cell division, and accumulation of lamin A within the nucleolus. Our work suggests that a subset of proteins reported as ubiquitin targets may instead be modified by ubiquitin variants that are the products of wrongly annotated pseudogenes and induce different functional effects.

[1] Department of Immunology and Cell Biology, Sherbrooke, QC, Canada. [2] Department of Biochemistry and Functional Genomics, Sherbrooke, QC, Canada. ✉email: xavier.roucou@usherbrooke.ca; fm.boisvert@usherbrooke.ca

Although it was initially described as a mechanism targeting proteins for degradation mediated by the proteasome[1], ubiquitylation can also regulate protein localization, protein–protein interactions as well as other protein functions[2]. Ubiquitylation is the covalent addition of the small 76 amino acids ubiquitin protein (Ub) via an isopeptide linkage between the C-termini diglycine (GG) motif of Ub and the NH2 group of lysine residues on the target protein[1]. The enzymatic reaction resulting in the attachment of the Ub involves a sequence of three reactions mediated by E1 activating, E2 conjugating, and E3 Ub ligase enzymes, the later determining the substrate specificity[3,4]. In humans, there are 2 E1 ligases, 35 E2 ligases, and over 600 predicted E3 ligases[5], underlining the complexity of this posttranslational modification and the specificity mediated by the E3 ligases.

Monoubiquitylation or multimonoubiquitylation on a single protein typically regulates intra- and intermolecular interactions[6]. For example, monoubiquitylation of proliferating cell nuclear antigen (PCNA) on Lys164 in response to a genotoxic stress promotes the recruitment of DNA polymerases lacking proofreading activity to overcome a DNA replication block and bypass DNA lesions[7]. Ub itself contains seven modifiable lysine residues, and multiple cycles of ubiquitination can also occur to produce long poly-Ub chains[8]. These poly-Ub chains, when attached to a target protein, generates a complex Ub code which stores different information depending on the Lys residue of the Ub that is linked[6]. For example, Lys-11-linked is involved in ERAD (endoplasmic reticulum-associated degradation) and in cell-cycle regulation, while Lys-48-linked is involved in protein degradation via the proteasome[9]. In addition to these homogenous chains, the Ub code also includes mixed and branched chains composed of heterogeneous linkages[6].

Ubiquitylated targets are recognized by Ub-binding domains (UBDs) present in proteins which transmit the information coded by the modification type to modulate various cellular processes[10]. The propensity of UBDs for monoubiquitylation or chains of specific length and linkage is an essential feature for their specificity of action[11]. Similar to Ub modifications, Ub-like modifiers (UBLs) proteins, such as SUMO or NEDD8, have their own specific E1 (activating), E2 (conjugating), and E3 (ligating) enzymes[9]. There is also cross-regulation between the various conjugation pathways, since some proteins can become modified by more than one Ub-like modifier, and sometimes even at the same lysine residue[12]. While UBLs share little amino acid identity with Ub (SUMO has only 18% amino acid identity with ubiquitin), their three-dimensional structures are virtually superimposable[13].

In human, Ub is encoded by four different genes: the UBA52 and RPS27A genes code for a single copy of Ub fused into the ribosomal proteins L40 and S27A, respectively[14,15]. UBB and UBC genes code for poly-Ub precursors (UBB encodes for three repeats and UBC for nine repeats in human)[16]. These precursors are processed into single Ubs by deubiquitinating enzymes (DUBs), proteases involved in removing Ub from ubiquitylated proteins, as well as cleaving Ub precursors[17]. Additional genes potentially coding for Ub have been identified, but are labeled as pseudogenes[18]. A pseudogene is defined as a copy of a gene that has lost the capacity to produce a functional protein[19]. Because of the similarities with the known Ub producing genes, those pseudogenes are assumed to be the products of integrations of reverse transcribed mRNAs, or arose from crossing over of ancestral alleles. The accepted conclusion is that they are either not transcribed or that they do not encode functional proteins, yet, ribosome profiling and proteogenomics approaches with customized protein databases including unannotated, pseudogenes, or alternative open reading frames (ORFs) recently detected these proteins[20–25]. Importantly, these proteins are usually not accounted for in current protein databases and, as such, are usually not identified in proteomic studies[26].

Here, we show that a subset of Ub pseudogenes are expressed, and are functionally different from canonical Ub. They are attached to different protein targets and do not appear to specifically target proteins for proteasomal degradation. Proteins modified by this Ub variant include lamins, and knockout (KO) cells display a nucleolar accumulation of lamins and slower cell growth.

## Results

**UBB pseudogene expression**. RNA-sequences annotated as noncoding are increasingly detected as translated and functionally characterized[27,28]. In particular, ribosome profiling and mass spectrometry (MS)-based proteogenomics confirmed that several pseudogene-derived long noncoding RNAs are the important sources of ORFs[22,25,29]. Yet, pseudogene function at the protein level is rarely considered[30]. In human, there are four Ub genes, UBB, RPS27A, UBA52, and UBC, as well as at least 52 pseudogenes of these genes (Supplementary Table 1). While over half of Ub pseudogenes-derived RNAs have not been detected, some can be detected in tissues alongside Ub RNAs (Fig. 1a and Supplementary Figs. 1–3). This suggests that at least some of those pseudogenes could potentially result in mRNA translation and production of proteins. In particular, UBB pseudogene 4 (UBBP4), RPS27A pseudogene 16 (RPS27AP16), and UBA52 pseudogene 8 (UBA52P8) are ubiquitously expressed at relatively high levels in several different datasets (Illumina Body Map, GTEx Portal and NIH Consortium) (Supplementary Figs. 1–3)[31].

UBB has five pseudogenes called UBB pseudogenes 1–5, or UBBP1–5[32,33] (Fig. 1b). UBBP1, UBBP2, UBBP3, and UBBP5 are processed pseudogenes resulting from reverse transcribed mRNA integration into the genome[34], but UBBP4 contains an intron and lacks a poly-A tail reminiscent of an integrated reverse transcript and is thus annotated as a transcribed unprocessed pseudogene[35]. Interestingly, evidence of expression of proteins encoded by UBBP4 were reported and we sought to elucidate these observations[29,36]. Because of the presence of a premature stop codon within the first Ub repeat in addition to having a frameshift in the third, it was concluded that UBBP4 was unproductive (Fig. 1b)[32]. However, since isolation and early sequencing of that cDNA back in 1988, its sequence within the current consensus human genome differs and does not contain the stop codon reported within the first Ub subunit, indicating that the mRNA can produce a poly-Ub precursor. The sequencing of PCR products from both genomic DNA and reversed transcribed mRNA confirmed the absence of the stop codon within the first ubiquitin repeat of UBBP4, as well as the transcription and splicing of the intron in three different commonly used cell lines (HEK293, HeLa, and U2OS) (Supplementary Fig. 4). Analysis of RNA-Seq data from ENCODE from nine different cell lines also indicates transcription at both exons (Supplementary Fig. 5)[37], and the presence of elongating ribosomes from ribosome profiling studies (Supplementary Fig. 6)[38] suggests that the mRNA produced from UBBP4 is actively translated into proteins. These data confirm that UBBP4 is ubiquitously transcribed and that the mRNA is actively translated into proteins by ribosomes.

**UBBP4 encodes four different Ub variants**. UBBP4 displays two ORFs (Figs. 1b and 2a). The first ORF includes three potential Ub variants termed Ubbp4^{A1}, Ubbp4^{A2}, and Ubbp4^{A3}. Ubbp4^{A1} differs from the canonical Ub by eight amino acids, while Ubbp4^{A2} has only one amino acid substitution, from a threonine to a serine (Fig. 2a). Ubbp4^{A3} cannot be a functional variant as it contains a frameshift and lacks the C-terminal diglycine required for attachment to other proteins (Fig. 2a). The second ORF

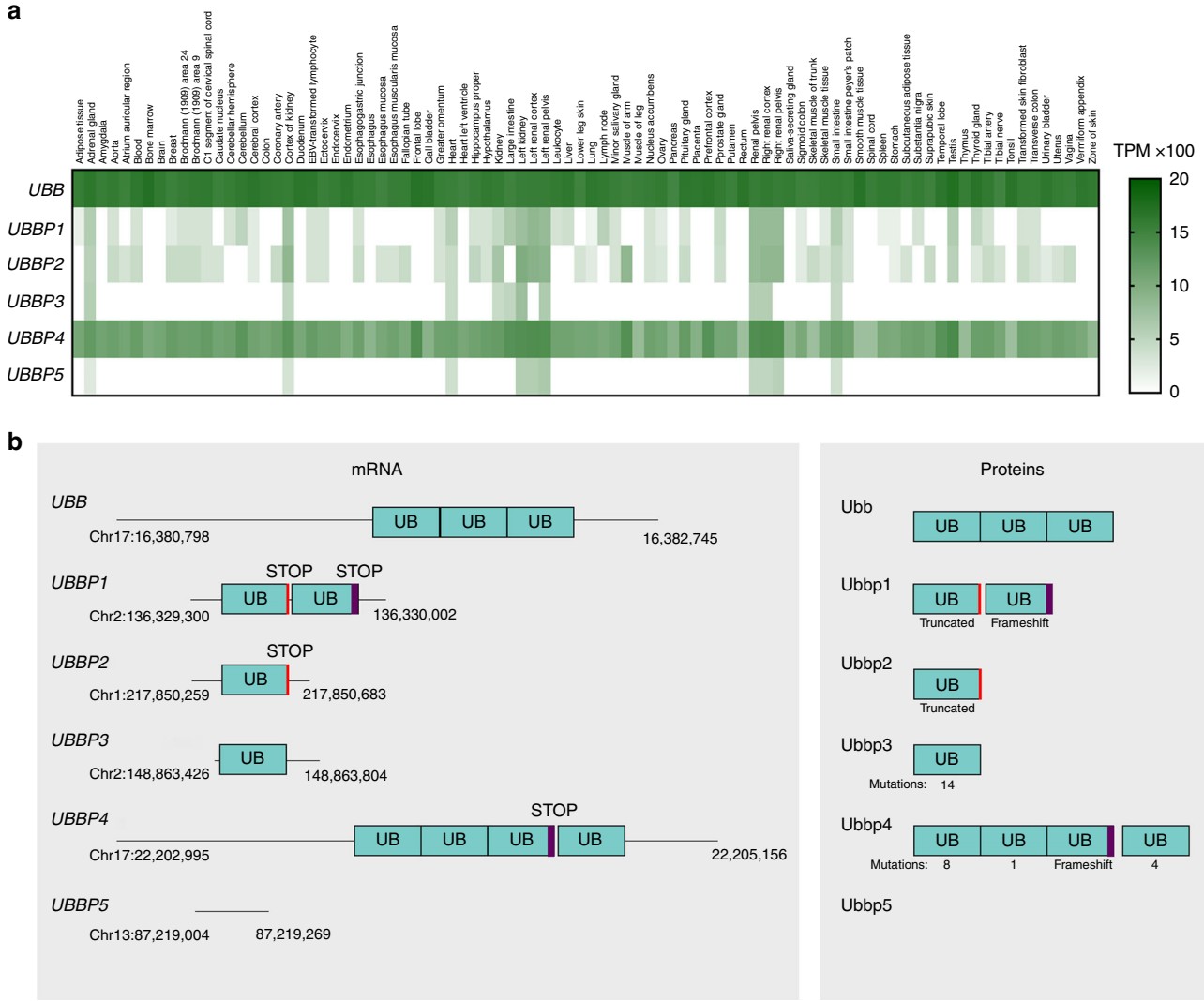

**Fig. 1 UBB pseudogenes are expressed and encodes ubiquitin variants. a** Expression levels analysis of *UBB* and *UBB* pseudogenes 1–5 through RNASeq data available through GTEx Portal from different tissues. **b** Schematic of *UBB* and *UBB* pseudogenes 1–5 mRNA and proteins potentially resulting from translation. Only *UBBP3* and *UBBP4* encode Ub variants with a diglycine at the C-terminal. The *UBBP4* gene contains two ORFs. The first one encodes three Ub repeats, with the third Ub containing a frameshift resulting in a premature stop codon. The second ORF encodes a Ub monomer.

contains another variant, Ubbp4^B1 (Figs. 1b and 2a). Re-analysis of MS data with the proteogenomics database OpenProt[36] identifies unique peptides matching to Ubbp4^A1, Ubbp4^A2, and Ubbp4^B1 in several proteomics studies[39–46], confirming the translation of these Ub variants (Fig. 2b). Interestingly, those unique peptides are also observed in interactome MS-studies (Fig. 2b)[40,42,43], suggesting that proteins can be modified by *UBBP4*-derived Ub variants.

**Ub variants are conjugated to protein targets**. To determine whether these Ub variants can be attached to other proteins, plasmids expressing HA-tagged Ubbp4^A1, Ubbp4^A2, Ubbp4^A3, and Ubbp4^B1 were transfected in cells, and whole cell lysates were separated by SDS-PAGE and immunoblotted with an HA antibody (Fig. 2c). Ubbp4^A2 and Ubbp4^B1 transfected cells display a large number of high-molecular weight proteins recognized by the HA antibody, similar to that observed with cells transfected with Ub. In contrast, Ubbp4^A1 and Ubbp4^A3, display only the monomer (unattached) (Fig. 2c), suggesting that the L73R change in Ubbp4^A1 could affect the ability to be activated by the E1, and that the lack of the diglycine at the C-terminal of Ubbp4^A3

prevents the covalent addition of ubiquitin to other proteins (Fig. 2a). Treatment of cells with the proteasome inhibitor MG132 results in the accumulation of higher molecular weight proteins in cells expressing Ub, as well as Ubbp4^A2, but not Ubbp4^B1 (Fig. 2d). Interestingly, while the cellular localization of Ub and Ubbp4^B1 appears to be both nuclear and cytoplasmic, Ubbp4^B1 shows a more prominent nuclear localization (Fig. 2e). Moreover, treatment with MG132 results in the accumulation of ubiquitin-modified proteins within cytoplasmic foci reminiscent of aggregates (Fig. 2e), and this accumulation is not observed with Ubbp4^B1. These observations indicate that proteins modified with ubiquitin accumulate when the proteasome is inhibited, consistent with one of the main function for ubiquitin modification. It also support the observation that proteins modified by Ubbp4^B1 are not targeted for proteasomal degradation. These results suggest that protein modification by Ubbp4^B1 has different faith and functions.

**Proteomic identification of Ub^KEKS targets**. *UBBP4* thus encodes a variant of Ub (Ubbp4^B1), which we proposed to name Ub^KEKS, because it differs from the canonical ubiquitin by four

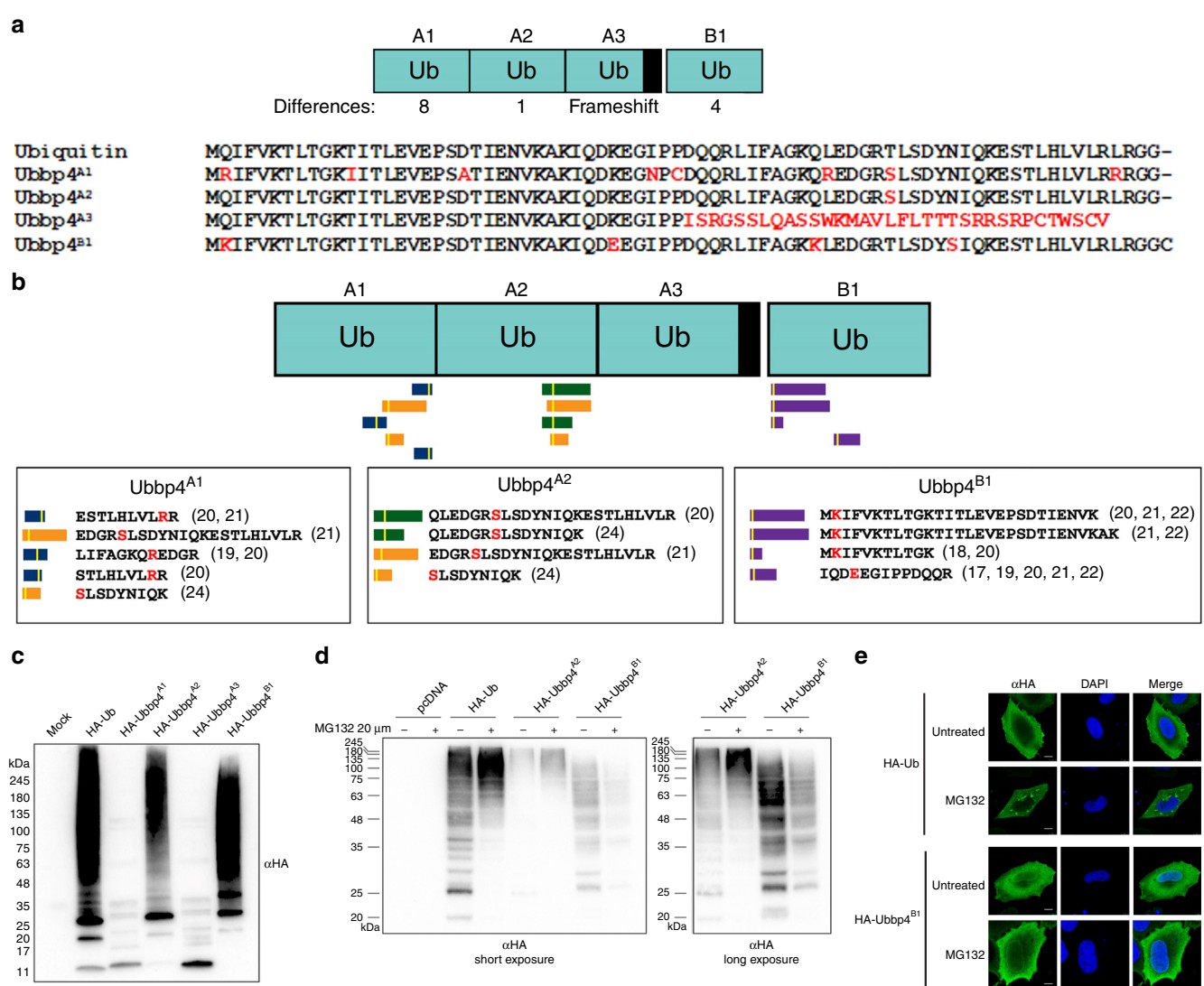

**Fig. 2 *UBBP4* encodes a functional ubiquitin variant without proteasomal targeting. a** The *UBBP4* gene encodes three ubiquitin repeats within a first ORF, Ubbp4[A1], Ubbp4[A2], and Ubbp4[A3], and a fourth ubiquitin within a second ORF, Ubbp4[B1]. The differences in amino acids compared with Ub are highlighted in red. **b** Identification of unique peptides in different large-scale proteomics experiments. The white bars in the peptides indicates the approximate position of the amino acids differences with the canonical Ub, and are shown in red. The unique peptides identified are shown with the reference to the datasets. **c** Total cell extracts of HeLa cells transfected with HA-tagged Ub, or Ubbp4[A1], Ubbp4[A2], Ubbp4[A3], or Ubbp4[B1] were separated by SDS-PAGE and revealed with an HA antibody (*n* = 3 biologically independent experiments). **d** HeLa cells were either untransfected, or transfected with plasmids expressing HA-tagged Ub or Ubbp4[A2] or Ubbp4[B1], and treated or not with MG132. Total cell extracts were separated by SDS-PAGE, and proteins were revealed by immunoblotting with an HA antibody (*n* = 3 biologically independent experiments). **e** HeLa cells transfected with either HA-Ub or HA-Ubbp4[B1] were treated or not with MG132 overnight and labeled for immunofluorescence microscopy with an HA antibody. The nuclei were stained with DAPI (*n* = 3 biologically independent experiments). Scale bars indicate 10 μm. Source data are provided as a Source Data file.

amino acids: Q2K, K33E, Q49K, and N60S (Fig. 2a). More importantly, this variant holds potential for different ubiquitin chains (K2 and K49), as well as missing K33. Because K49 is located adjacent to the major proteasomal degradation signal (poly-K48 chains)[47], this suggests a mechanism of modification that could explain the lack of protein degradation of proteins modified by Ub[KEKS]. The expression of HA-tagged Ub[KEKS] results in its covalent attachment onto other proteins as revealed by immunoblotting, and interestingly, the pattern of proteins modified is different when compared with Ub (Figs. 2d and 3a). In order to identify proteins modified by or interacting with Ub[KEKS], tetracyclin inducible stable cell lines expressing either HA-tagged Ub, or HA-tagged Ub[KEKS] were used for affinity purification, followed by MS-based protein identification. These experiments were performed in either denaturing or

nondenaturing conditions. The rationale was to differentiate the identification of proteins directly modified by Ub or Ub[KEKS], and to identify proteins interacting with these small modifiers. In order to accurately define Ub[KEKS] specific modifications compared with Ub, SILAC-labeled cells were used to compare non-modified (light), Ub-modified (medium), and Ub[KEKS]-modified (heavy) proteins (Fig. 3b). Mixing the immunoprecipitates prior to MS identification allows to quantify enrichment of proteins in either the Ub or Ub[KEKS] expressing cells, in relation to control cells and with each other (Fig. 3b). Proteins identified as using HA-Ub versus HA-Ub[KEKS] confirmed our initial observation by immunoblotting that the protein targets of Ub[KEKS] are different from targets of Ub, demonstrating a specificity in protein substrates (Fig. 3c, d and Supplementary Data 1). Moreover, under nondenaturing conditions, most subunits of the proteasome were

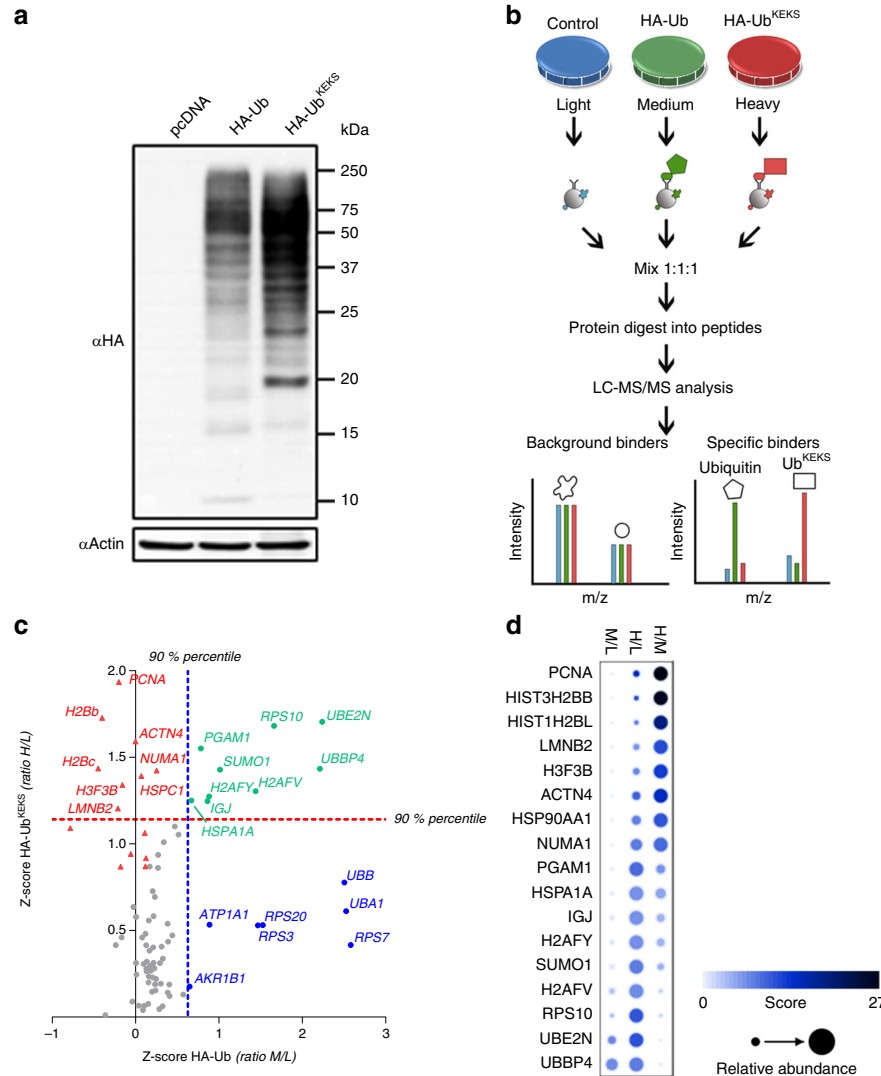

**Fig. 3 Ub^KEKS targets proteins with a different specificity compared with Ub. a** HeLa cells were transfected with empty vector (pcDNA), HA-Ub or HA-Ub^KEKS. Proteins were extracted and analyzed by western blot with antibodies against the HA tag or actin ($n = 3$ independent experiments). **b** Proteins from SILAC-labeled cells either with light amino acids (control) or with heavier isotopes (M or L) for quantification of proteins enriched in HA-Ub IPs versus HA-Ub^KEKS IPs. **c** Z-scores of H/L and M/L ratios for each protein detected in all conditions (heavy, medium, and light). Proteins significant (above the 90% percentile) for HA-Ub are indicated in blue, in red for HA-Ub^KEKS, and in green for both. Significance lines are indicated as a blue dotted line (90% percentile for M/L z-scores) and as a red dotted line (90% percentile for H/L ratios). Proteins with a significantly high H/M ratio (specific to Ub^KEKS versus Ub) are indicated as red triangles. Axis are limited to area of interest, thus some values do not appear on this graph. These experiments were performed in biological triplicates ($n = 3$). **d** Proteins identified in at least two replicates with a significant ratio of enrichment are represented using a dot plot. The color and size of the dots are proportional to their z-scores and relative abundance, respectively. Source data are provided as a Source Data file.

enriched only with Ub, confirming that Ub targets proteins to the proteasome, but not Ub^KEKS (Supplementary Fig. 7 and Supplementary Data 1).

**Lamins are modified by Ub^KEKS.** Amongst the proteins identified specifically with Ub^KEKS are the nuclear lamins (lamin A, B1, and B2) (Fig. 3c, d and Supplementary Data 1), which constitute the major architectural proteins of the nuclear lamina[48]. Mutations in lamins cause a number of related genetic diseases termed laminopathies. Ubiquitylation of lamins may contribute to the regulation of these mechanisms involved in causing these diseases[49,50]. Interestingly, the nuclear localization of Ub^KEKS appears more intense at the nuclear periphery, consistent with lamins being a major target for modification by Ub^KEKS (Fig. 2e). Co-immunoprecipitations of lamin A or B2 with either HA-

tagged Ub, or HA-tagged Ub^KEKS confirms that lamin A and B2 are modified preferentially by Ub^KEKS when compared with Ub (Fig. 4a, b). To determine whether Ub^KEKS is essential for cell function, KO of *UBBP4* was performed using a CRISPR/Cas9 approach with paired guide RNAs designed to delete 715 bp (strategy 1) or 1659 bp (strategy 2) (Fig. 4c and Supplementary Table 2). Clones were obtained from both strategies showed a significant delay in cell growth (Fig. 4d). Interestingly, *UBBP4* KO cells also showed an accumulation of lamin A within nucleoli compared with WT cells (Fig. 4e), indicating that modification of lamin A by Ub^KEKS could be involved in regulating its subcellular localization. The nucleoli of *UBBP4* KO cells are also enlarged (Fig. 4f, g), suggesting that accumulation of lamin A could interfere with nucleolar function, such as ribosome biogenesis. The localization of lamin A in the nucleolus observed in the KO cells, as well as the size of the nucleolus, was rescued 72 h

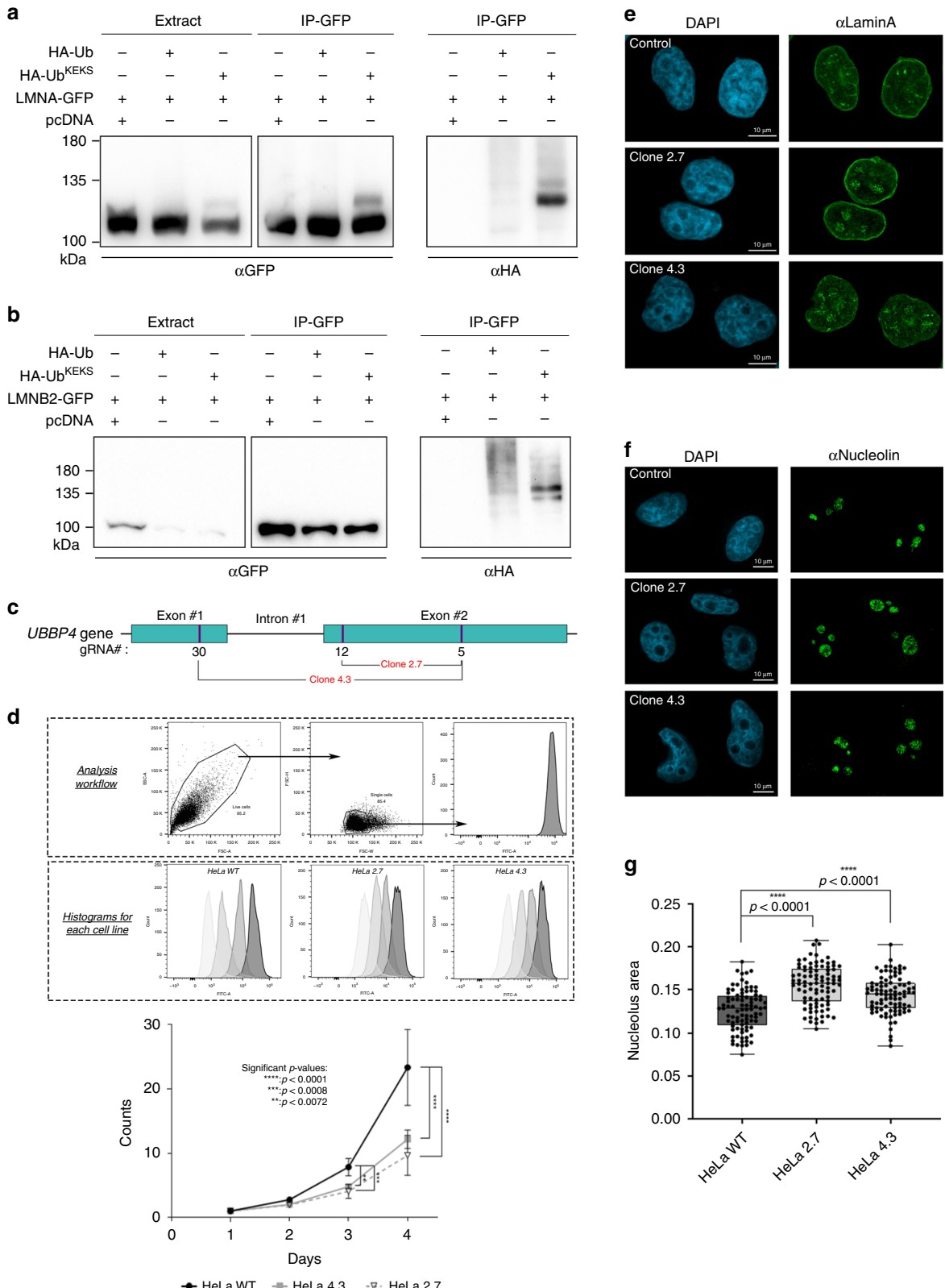

following transient transfection of a plasmid encoding HA-Ub$^{KEKS}$ (clone 4.3 and 2.7) (Supplementary Fig. 8, compare transfected versus nontransfected cells).

**Absolute quantification of Ubiquitin and Ub$^{KEKS}$.** Absolute quantification of Ubiquitin and Ub$^{KEKS}$ was performed using the

AQUA method[51]. There are four unique peptides for Ub$^{KEKS}$ that have been detected in several large-scale studies (Fig. 2b)[39–46]. Of those, three have miscleavages and included the N-terminal methionine which is often found modified (see peptide sequences in Fig. 2b). This left only one peptide that had no miscleavage, no internal lysine or arginine and was detected in five independent

**Fig. 4 UbKEKS is important for cell growth and regulates lamin A localization. a–b** Cells were transfected with either GFP-tagged lamin A (LMNA) or lamin B2 (LMNB2), and with HA-tagged Ub or UbKEKS. Total cell lysates (Extract) or IPs with GFP-Trap were loaded on SDS-PAGE and revealed with GFP or HA antibodies ($n = 3$ independent experiments). **c** Knockout of *UBBP4* was performed using CRISPR/Cas9 and two different combinations of guide RNAs in HeLa cells. **d** Cells are sorted from the debris according to FSC-A and SSC-A, and then single cells are selected using FSC-H and FSC-W for measuring using the FITC channel. Wild type and KO HeLa cells (clones 2.7 and 4.3) were assessed for growth using a CFSE assay at each time points for 96 h. Data are presented as mean values with ±SD. Statistical analysis was performed using two-way ANOVA ($F(2,84) = 45.70$) and Dunnett's multiple comparison post hoc tests ($n = 8$ independent experiments). **e** Wild type and KO HeLa cells (clones 2.7 and 4.3) were labeled for immunofluorescence microscopy with a lamin A antibody. The nuclei were stained with DAPI ($n = 3$ independent experiments). **f** Control and KO HeLa cells (clones 2.7 and 4.3) were labeled for immunofluorescence microscopy with a Nucleolin antibody. The nuclei were stained with DAPI ($n = 3$ independent experiments). **g** Quantification of nucleoli was achieved using CellProfiler, through a primary object identification using DAPI and Nucleolin staining, respectively. Each nucleolus was reported to the parental nucleus and the area was determined by measuring the object size shape. Data are presented as box plots where the center lines show the medians; box limits indicate the 25th and 75th percentiles as determined by Prism software; whiskers extend to minimum and maximum values; individual data points are superimposed on the box plot. Statistical analysis was performed using two-way ANOVA ($F(2,540) = 37.08$) and Dunnett's multiple comparison post hoc tests ($n = 90$ cells examined over three independent experiments). Source data are provided as a Source Data file.

studies. We analyzed the product ions of the synthetic peptides to identify which ion showed the highest intensity (Supplementary Fig. 9a). While the y6++, y5, and y6 ion intensities were relatively high for Ub, only the y6 seemed suitable for quantification for UbKEKS. To be consistent, we chose one ion for each of Ub and UbKEKS for quantification. In some of the quantification, an early eluting peak was observed in presence of the cell extract (Fig. 5 and Supplementary Figs. 9 and 10). This peak is not observed when the heavy peptide is analyzed alone, and is only present when a cell extract is added (Supplementary Fig. 9b). This was confirmed to not be related to UbKEKS, as the signal of that early eluting peak did not increase when analyzing a whole cell extracts from cells over-expressing UbKEKS (Supplementary Fig. 9b), while the signal for UbKEKS co-eluting with the heavy peptide did increase. This confirmed that the early eluting peak is a contaminant peak originating from the cell extracts, but it is not UbKEKS.

To measure the amount of ubiquitin and UbKEKS, exponentially growing HeLa WT were harvested and lysed. Following tryptic digestion, the Heavy Arginine (U-13C6, 15N4; mass difference: +10 Da) labeled AQUA Ub (EGIPPDQQR) and UbKEKS (IQDEE-GIPPDQQR) peptides were spiked into the samples at a final concentration of 1.66 fmol/μl. The peptides were analyzed by a mass spectrometer using a PRM method with an inclusion list containing the $m/z$ values corresponding to the monoisotopic form of the heavy and light peptides of Ub (520.2/525.2) and UbKEKS (762.8/767.8). For quantification, the most intense fragment ion (y6) was used for both peptides (Fig. 5a and Supplementary Fig. 9a). The amount of Ub and UbKEKS proteins were calculated using the light to heavy peptide ratio and measured according to the concentration of spiked in AQUA peptides. The concentrations of Ub measured was 142 fmol/μg of total protein (Supplementary Fig. 10 and Supplementary Data 2), which is consistent with previous reports measuring Ub with a similar approach (99 fmol/μg in HeLa, 115 fmol/μg in HEK293 and 111 fmol/μg in HCT116)[52,53]. The concentration of UbKEKS in cells as expected was significantly lower, in the order of 0.23 fmol/μg. By comparison, quantification of Nedd8, Sumo1, and Sumo2 using a similar approach is 11, 0.7, and 20 fmol/μg, respectively[53].

**Absolute quantification of Ub and UbKEKS on lamins**. To measure the amount of endogenous Ub and UbKEKS on lamin A and lamin B2, wild type and UbKEKS KO HeLa cells (clone 4.3) were transfected with either GFP-lamin A or GFP-lamin B2, and the GFP-tagged proteins were immunoprecipitated. Following digestion, AQUA Ub and UbKEKS peptides were spiked into the samples. The peptides were analyzed by a mass spectrometer using the same PRM method as described above (Fig. 5b). The ratio of UbKEKS to Ub on lamin A and lamin B2 were calculated (Fig. 5c and Supplementary Data 3). Interestingly, the ratio

measured an ~20:1 ratio of Ub:UbKEKS on both lamin A and lamin B2, suggesting that modification of lamins remains majoritarily by endogenous Ub instead of UbKEKS, although the ratio is much higher compared with the 700:1 ratio measured in whole cell extracts. The amount of UbKEKS decreased significantly in the KO cells, further validating the CRISPR/Cas9-mediated KO of the UBBP4 gene expression (Fig. 5c and Supplementary Data 3). Because UbKEKS was quantified on shorter gradients in order to increase the peak intensity of UbKEKS by reducing the peak width, we did have a slight overlap in signal from the early eluting contaminant, which does provide some signal when performing the quantification (Fig. 5 and Supplementary Fig. 9c). As demonstrated on the LMNA pulldown experiments comparing the WT and KO cells, we can see that the peak corresponding to UbKEKS in the WT cells is perfectly co-eluting with the AQUA peptide (black arrow), and this peak is not observed in the KO cells (Supplementary Fig. 9c). However, the quantification does not measure a complete KO of UbKEKS, because we did have some signal trailing from that early eluting peak (see the red trail left in the HeLa 4.3, Supplementary Fig. 9c). We can thus conclude that cells that are KO for UbKEKS are not likely to have remnants of UbKEKS, which was also confirmed by genomic sequencing.

## Discussion

Here, we establish that UBBP4 is a genuine protein-coding gene with two functional ORFs encoding Ub variants. In particular, our results strongly suggest that UbKEKS does not target proteins for degradation by the proteasome, but modifies proteins, including lamins which localization are different in the absence of UbKEKS. The identification and demonstration that a pseudogene for ubiquitin can encode functional modifiers that are slightly different obviously raises several interesting biochemical questions about UbKEKS. Engineered ubiquitin variants have been developed as probes to determine interaction with components of different proteins involved in the ubiquitin system[54]. These mutagenesis studies revealed several substitution that were important for interaction with UBDs, but also to generate potent inhibitors of E3 ligases and DUBs[55,56]. Interestingly, while the residues that are different between Ub and UbKEKS do not appear critical for the HECHT or RING E3 ligases, residues 2 and 49 are important for USP binding[57], which could suggest partial resistance to Ub-specific proteases.

Considering the breadth of functions associated with the different polyubiquitin chains, the presence of two additional lysines (K2 and K49) raises the possibility of chains with different functions. It is possible to identify sites of modification of ubiquitin by analyzing the diglycine remnant on lysines following trypsin digestion by MS. Analysis of the immunoprecipitates followed by MS identification of proteins in cells transfected with

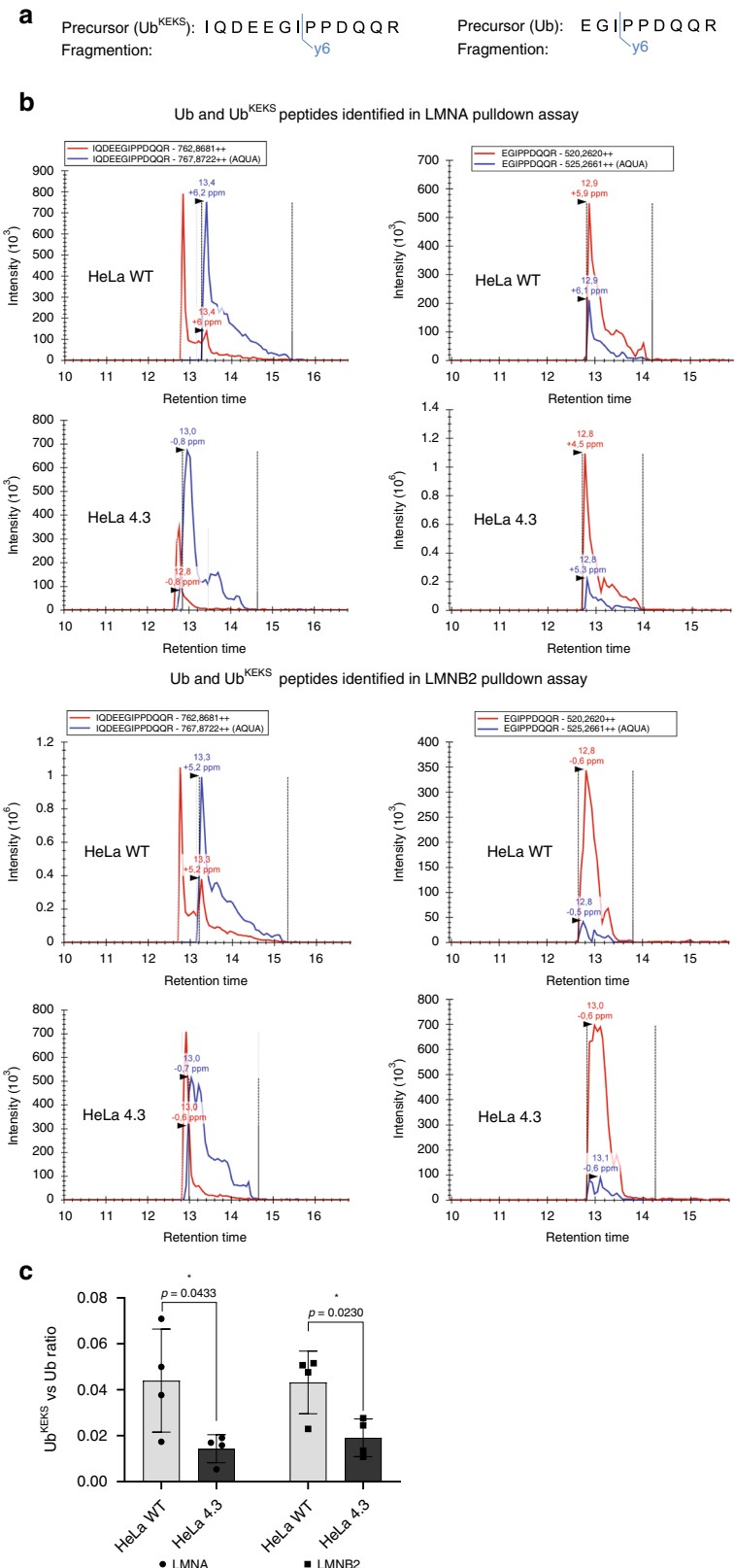

**Fig. 5 Quantification of Ub and Ub$^{KEKS}$ in lamin A and lamin B2 pulldown assay. a** Precursors and their fragment ions used for quantification of Ub and Ub$^{KEKS}$. **b** Fragment ion chromatogram of Ub and Ub$^{KEKS}$ in WT and Ub$^{KEKS}$ KO (clone 4.3) HeLa cells in lamin A (LMNA) and lamin B2 (LMNB2) pulldown assays. Blue line represents heavy (AQUA) peptide, red line represents endogenous peptides with *m/z* values indicated in the box above chromatograms. Dotted lines indicate peak boundaries with black arrows showing peaks. Peak values with mass error are indicated above each peak. **c** Ratio of Ub$^{KEKS}$ to Ub in WT and Ub$^{KEKS}$ KO (clone 4.3) HeLa cells in LMNA and LMNB2 pulldown assays. Data are presented as mean values ± SD; individual data points are superimposed on the bar graph. Statistical analysis was performed using unpaired two-tailed *t*-test (*n* = 4 independent experiments). Source data are provided as a Source Data file.

Ub$^{KEKS}$ using diglycine modification of lysines as a variable modification did identify K49, as well as K48/K49 modified Ub$^{KEKS}$-specific peptides, supporting the formation of different ubiquitin chains (see peptides file, PRIDE repository). Interpretation of the signals encoded by ubiquitylation depends on the recognition of mono-Ub or Ub-chains through UBDs[10,11,58]. The observation that Ub$^{KEKS}$ does not promote proteasomal degradation like Ub may be the result of a loss of recognition by the proteasome receptors, the presence of K49, or chains forming on K49. Another interesting observation is the possibility that Ub$^{KEKS}$ acts as a Ub-chain terminator like SUMO1[59]. This could be consistent with the much lower amount of Ub$^{KEKS}$ compared with Ub (approximately a 700-fold difference), which is also the case for SUMO1 compared with SUMO2 and SUMO3, but to a lesser extent[53]. This is suggested by the presence of more defined Ub$^{KEKS}$ chains at +1 and +2 on lamins compared with Ub, which presents higher molecular weights modifications indicative of poly-Ub chains (Fig. 4a, b).

The specificity for target proteins modified by Ub$^{KEKS}$ compared with Ub as demonstrated by the different protein profiles seen by immunoblotting of cell extracts transfected by both (Fig. 2c, d), as well as by identification of modified proteins by MS (Fig. 3), suggest that the machinery involved in the modification can differentiate between the two ubiquitin variants. It will certainly be interesting to identify whether some E2 or E3 enzymes have different affinity for ubiquitin variants, and therefore can preferentially attach one of the variant instead of Ub. The large difference in stoichiometry between Ub and Ub$^{KEKS}$ would suggest that such an E2 or E3 would have a much higher affinity for Ub$^{KEKS}$, and it remains to be seen whether a subset of those enzymes are specific for Ub$^{KEKS}$, or whether a yet undiscovered domain will be identified. Several proteins modified by UbKEKS were identified (Fig. 3), such as lamins, PCNA and histones, but also several E2 and E3 ligases. For example, Ubr5, an E3 ligase, acts as a regulator of DNA damage response by acting as a suppressor of RNF168, an E3 ubiquitin ligase that promotes accumulation of Lys-63-linked histone H2A and H2AX at DNA damage sites. Interestingly, histone H2A was identified as one of the top candidates after PCNA and lamins in our SILAC AP-MS experiment (Fig. 3), suggesting a role in the regulation of recognition and signaling of DNA damage. Another E3 ligase identified, RNF138, has recently been involved in DNA repair[60]. In their study, the authors performed a MS experiment to identify proteins interacting with RNF138, and identified UBBP4 as their top hit (their Supplementary Data 1), suggesting that RNF138 could be an E3 ligase specific for Ub$^{KEKS}$.

The quantification of Ub and Ub$^{KEKS}$ on lamin A and B2 suggest that those proteins are modified mostly by ubiquitin by a ratio of 20:1. However, the amount of Ub$^{KEKS}$ is much higher on lamins when compared with the amount of Ub:Ub$^{KEKS}$ in cells which is 700:1, confirming that lamins can be specifically targeted for modification by Ub$^{KEKS}$. At least 20 ubiquitylation sites on lamin A have been identified in large-scale studies[61], and it is not possible to determine at this point whether one of them is a specific site for Ub$^{KEKS}$, or whether Ub$^{KEKS}$ competes with Ub for modification of lamins. Several E3 ligases have been identified that are able to modify lamins, including RNF123[49] and HECW2[50]. Overexpression of RNF123 results in delayed cell-cycle progression, and HECW2 has also been shown to target PCNA, which was also identified as a protein modified by Ub$^{KEKS}$.

The fact that the identification of diglycine residues on lysines by MS, and that antibodies recognizing ubiquitin are also likely cross-reacting with Ub$^{KEKS}$ suggest that other proteins reported as modified by Ub could be instead modified by Ub$^{KEKS}$ (Fig. 3c, d and Supplementary Data 1) as well as other ubiquitin variants (Supplementary Fig. 11). Indeed, at least five other Ub pseudogenes (UBA52P6,

RP11-367G18.2, UBBP3, CTD-3214H19, and RPS27AP1) have strong evidences at the mRNA or protein levels for the expression of unique ubiquitin variants, adding to the potential complexity of this modification. In addition to showing that the diversity of the Ub code has been underestimated, our results are a reminder that proteins coded in genes and transcripts originally annotated as noncoding remain to be discovered[36].

## Methods

**Sequencing of *UBBP4* from genomic DNA and mRNA.** Genomic DNA was extracted using the GenElute Mammalian Genomic DNA Miniprep Kit (Sigma-Aldrich), and RNA was isolated using TRIzol Reagent (Invitrogen) from one 100 mm petri cultures of 293, U2OS or HeLa cells. The isolated RNA was reversed transcribed using 1 μg of RNA and 50 μM oligo-dT with the ProtoScript II reverse transcriptase (New England Biolabs), and incubated at 42 °C for 60 min. The genomic DNA specific to *UBBP4* was amplified by PCR using a first oligonucleotide within the intron, and a second oligonucleotide 3′ of the exon 2 within the noncoding genomic region (see Supplementary Fig. 4) (ATCTAGGTCAAAAT GCGGATCTTCGT and ATGACAAAACACCAAGTATGCTACCATT). The amplified expected product of the genomic DNA is 1092 base pairs. The reversed transcribed mRNA including both exons was amplified using oligonucleotides from the 5′ to the 3′ untranslated regions of the mRNA (GCTGGTGCTGCAAGA AAGTT and TGCTACCATTCAACGAAACCT). The spliced sequence expected is 1456 base pairs.

Following PCR amplification, the products were separated on 1% agarose gels, and the DNA of the expected size were cut and extracted, followed by blunt cloning into pUC19 digested with SmaI. A blue-white colony selection with X-galactosidase was performed, and ten clones for each cell line were then sequenced (Université Laval Sequencing Platform).

**Analysis of expression from RNAseq data.** The Expression Atlas (http://www.ebi.ac.uk/gxa)[62] containing manually curated RNA-sequencing studies was queried using the human *UBA52*, *RPS27A*, and *UBB* genes as well as the annotated pseudogenes, with data from the GTEx Portal, the Illumina Body Map and the Human Protein Atlas. The expression profile across different tissues using the GTEx Portal data is displayed in transcripts per million or fragments per kilobase of exon model per million reads mapped in Fig. 1 and Supplementary Fig. 1.

**Mass spectrometry identification of *UBBP4* proteins.** Identification of unique peptides specific for pseudogenes or alternative ORFs was performed by re-analyzing previous proteomics experiments with an extended database including these additional putative proteins using available data from several large-scale studies. The information and the peptides identified can be searched in the database OpenProt, which includes all the details regarding the peptides identified[25]. Only unique peptides were considered valid for the identification of *UBBP4* proteins. The studies in which each of the unique peptides were identified are from several independent large-scale experiments[39–46].

**Cloning and plasmids.** HA-Ub or HA-Ub$^{KEKS}$ were synthesized as gblocks (Integrated DNA Technologies) and inserted into pcDNA 3.1 (Addgene) in XhoI-BamHI. Ha-Ub$^{KEKS}$ has the C-terminal cysteine, which is normally found on the UBBP4 gene. lamin A and B2 were amplified by PCR using oligonucleotides that included AttB recombination sites from a cDNA library generated by RT-PCR with an oligo-dT following isolation of mRNA by Trizol on U2OS cells (see Supplementary Table 2 for oligonucleotide sequences). The PCR product were then incorporated by recombination into pDONR 221 (Life Technologies) using BP recombinase and subsequently into pDEST47 using LR recombinase (Life Technologies).

**Cell culture, transfection, and immunoblotting.** HEK293, HeLa, or U2OS cells (ATCC CRL-1573, CCL-2, and HTB-96) were grown as adherent cells in Dulbecco's Modified Eagle Medium (DMEM) (Life Technologies) supplemented with 10% fetal bovine serum (Invitrogen), 100 U/ml penicillin/streptomycin and 2 mM GlutaMax. Cells were transfected with a control plasmid (pcDNA), or plasmids expressing HA-Ub or HA-Ub$^{KEKS}$ using Lipofectamine LTX (ThermoFisher Scientific), and treated or not with the proteasome inhibitor MG132 at a final concentration of 5 μM for 24 h. Cells were lysed directly in LDS sample buffer, resolved by SDS-PAGE, and transferred to nitrocellulose. Immunoblotting was performed with either a HA antibody (Invitrogen #26183, 1:1000)) or actin antibody (Sigma #A5441, 1:2000) or a GFP antibody (Roche #11814460001, 1:1000). Uncropped scans of all blots can be found in the Source Data file and in Supplementary Fig. 12.

**Co-immunoprecipitations.** Cells were transfected with GFP-LMNA or GFP-LMNB2 and a control plasmid (pcDNA 3.1), and plasmids expressing HA-Ub or HA-Ub$^{KEKS}$ using Effectene (QIAGEN). Following transfection, cells were harvested by scraping in PBS and lysed in high-salt (HS) buffer (1% NP-40, 50 mM Tris ph7.5, 300 mM NaCl, 150 mM KCl, 5 mM EDTA, 1 mM DTT, 10 mM NaF, 10%

glycerol, Roche Complete Protease Inhibitor Cocktail) for 10 min on ice. The lysates were then centrifuged for 10 min at $13,000 \times g$ at 4 °C and equal amount of proteins were incubated with GFP-Trap agarose beads from ChromaTek (Martinsried, Germany) for 2 h at 4 °C. Beads were then washed three times with HS buffer.

**CRISPR-Cas9-mediated UBBP4 KO.** CRISPR-Cas9-mediated *UBBP4* KO HeLa cells were generated according to[63] the minor modifications. Briefly, sgRNAs were designed using the Broad Institute sgRNA Designer (CRISPRko) tool (http://portals.broadinstitute.org/gpp/public/analysis-tools/sgrna-design)[64]. Good sgRNA candidates were further confirmed with the CCTOP tool (http://crispr.cos.uni-heidelberg.de/)[65] and were verified to not have predicted off target within *UBB*, *UBC*, *UBA52*, and *RPS27*, as well as other *UBB* pseudogenes. For sgRNA-sequences, see Supplementary Table 2. The sgRNA inserts were prepared by annealing the top and bottom oligos and cloned into the pSpCas9(BB)-2A-GFP plasmid (Addgene #48138, Cambridge, MA)[63]. The resulting plasmids were verified by sequencing. Two strategies using pairs of sgRNAs designed to excise the CDS of *UBBP4* were used (Fig. 4c). Enrichment for Cas9–2A-GFP expressing cells and isolation of clonal cell populations were performed 24 h after transfection by single-cell FACS sorting. Successfully edited clones KO for *UBBP4* were confirmed by PCR and sequencing. Genomic DNA was amplified with oligonucleotides spanning the edited/deleted genomic regions (Supplementary Data 2). Clones producing PCR products with the expected sizes according to the planned deletions were directle sequenced. CRISPR-ID[66] and TIDE[67] were then used to confirm the complete KO.

**Carboxyfluorescein succinimidyl ester cell growth assay.** The control and KO cells lines were labeled with the cell proliferation tracer carboxyfluorescein succinimidyl ester (CFSE) (Biolegend). A total of 125,000 cells were resuspended in 1 ml PBS and 1 ml of PBS containing 10 µM of CFSE was added. The cells were incubated at 37 °C for 20 min and the incorporation was stopped by adding 10 ml of complete DMEM as described above. Following centrifugation, the supernatant was removed and the cells were resuspended in complete DMEM, incubated for 10 min at room temperature and the cells were seeded into 60 mm dishes. At each time point (24, 48, 72, and 96 h), the cells were harvested using 500 µl of trypsin followed by two washes with ice-cold PBS. The cells were fixed in 100% ethanol and conserved at −20 °C. The quantification of CFSE was measured by flow cytometry (BD Fortessa cytometer, Becton Dickinson) and the analysis was achieved with the FlowJo software (LLC).

**Immunofluorescence.** Cells ($15 \times 10^3$) were seeded onto glass coverslips and grown for 24 h. Cells were rinsed twice with ice-cold PBS, fixed with 4% paraformaldehyde in PBS for 20 min on ice and washed twice with PBS. The cells were permeabilized with 0.15% Triton X-100 in PBS for 5 min and incubated with 10% goat serum in PBS for 20 min. The cells were then incubated in primary antibodies overnight for LMNA (Abcam #ab133256, 1:1000) and HA (Invitrogen #26183, 1:1000) or 2 h for Nucleolin (Abcam #ab136649, 1:2000) at room temperature in 10% goat serum in PBS. The cells were rinsed twice with PBS and incubated with AlexaFluor 488 goat anti-mouse (Invitrogen #A11001, 1:800) or anti-rabbit (Invitrogen #A11008, 1:800) at room temperature for 1 h. Following two PBS washes, the nuclei were stained by DAPI (1 µg/µl) for 10 min at room temperature, washed twice with PBS and mounted with Immuno-mount (ThermoFisher Scientific). The quantification of nucleoli area was achieved with CellProfiler (https://cellprofiler.org/). Briefly, the nuclei and nucleoli were identified through a primary object identification module using DAPI and Nucleolin staining, respectively. Each nucleolus was reported to the parental nucleus with the related objects module and the area was determine using the Measure object size shape module.

**Identification of Ub and Ub^KEKS modified proteins.** HeLa cells were grown in DMEM depleted of arginine and lysine (Life Technologies A14431-01) and supplemented with 10% dialyzed fetal bovine serum (Invitrogen, 26400-044), 100 U/ml penicillin/streptomycin and 2 mM GlutaMax. Arginine and lysine were added in either light (Arg 0, Sigma, A5006; Lys 0, Sigma, L5501), medium (Arg 6, Cambridge Isotope Lab (CIL), CNM-2265; Lys 4, CIL, DLM-2640), or heavy (Arg 10, CIL, CNLM-539; Lys 8, CIL, CNLM-291) form to a final concentration of 28 µg/ml for arginine and 49 µg/ml for lysine. L-proline was added to a final concentration of 10 µg/ml to prevent arginine to proline conversion. Cells grown in each SILAC medium were transfected with a control plasmid (pcDNA), or plasmids expressing HA-Ub or HA-Ub^KEKS using GeneCellin (Bulldog Bio). Following transfection, cells were harvested separately by scraping in PBS, and then lysed in HS buffer (1% NP-40, 50 mM Tris ph7.5, 300 mM NaCl, 150 mM KCl, 5 mM EDTA, 1 mM DTT, 10 mM NaF, 10% glycerol, Roche Complete Protease Inhibitor Cocktail) for 10 min on ice. Alternatively, cells were lysed in denaturing buffer (2% SDS, 150 mM NaCl, 10 mM Tris-HCl, pH 8.0 with 2 mM sodium orthovanadate, 50 mM sodium fluoride, and Roche Complete Protease Inhibitor Cocktail), incubated at 95 °C for 10 min, sonicated and diluted with nine volumes of dilution buffer (10 mM Tris-HCl, pH 8.0, 150 mM NaCl, 2 mM EDTA, 1% Triton). The lysates were then centrifuged for 10 min at $13,000 \times g$ at 4 °C and equal amount of proteins were incubated with an HA antibody (12CA5 monoclonal antibody, Millipore Sigma #11583816001) for 2 h at 4 °C. Beads were then washed three times with IP buffer and two times with PBS. After the last wash, the beads from the three SILAC

conditions were resuspended in PBS and combined before removing the remaining PBS. The beads were then resuspended in sample buffer and processed for on-beads digestion. For each immunoprecipitations, proteins on beads were reduced in 10 mM DTT, boiled and alkylated in 15 mM iodoacetamide. The proteins were digested overnight with trypsin (Trypsin Gold, Mass Spectrometry Grade, Promega Corporation, WI, USA). The resulting tryptic peptides were extracted by 1% formic acid, then 60% $CH_3CN$/0.1% formic acid, lyophilized in a speedvac, and resuspended in 1% formic acid. These experiments were performed in biological triplicates ($n = 3$).

**SILAC-based quantitative proteomics.** Trypsin-digested peptides were loaded and separated onto a nanoHPLC system (Dionex Ultimate 3000). A total of 10 µl of the sample (2 µg) was first loaded with a constant flow of 4 µl/min onto a trap column (Acclaim PepMap100 C18 column, 0.3 mm id × 5 mm, Dionex Corporation, Sunnyvale, CA). Peptides were then eluted off towards an analytical column heated to 40 °C (PepMap C18 nano column (75 µm × 50 cm)) with a linear gradient of 5–35% of solvent B (90% acetonitrile with 0.1% formic acid) over a 4 h gradient at a constant flow (200 nl/min). Peptides were then analyzed by an OrbiTrap QExactive mass spectrometer (Thermo Fischer Scientific Inc.) using an EasySpray source at a voltage of 2.0 kV. Acquisition of the full scan MS survey spectra ($m/z$ 350–1600) in profile mode was performed in the Orbitrap at a resolution of 70,000 using 1,000,000 ions. Peptides selected for fragmentation by collision-induced dissociation were based on the ten highest intensities for the peptide ions from the MS survey scan. The collision energy was set at 35% and resolution for the MS/MS was set to 17,500 for 50,000 ions with maximum filling times of 250 ms for the full scans and 60 ms for the MS/MS scans. All unassigned charge states as well as singly, seven and eight charged species for the precursor ions were rejected, and a dynamic exclusion list was set to 500 entries with a retention time of 40 s (10 ppm mass window). To improve the mass accuracy of survey scans, the lock mass option was enabled. Data acquisition was done using Xcalibur version 2.2 SP1.48. Identification and quantification of proteins identified by MS were done using the MaxQuant software version 1.5.2.8[68]. Biological replicates were done three times and combined together for the MaxQuant analysis. Quantification was done with light (Lys 0 and Arg 0), medium (Lys 4 and Arg 6), and heavy (Lys 8 and Arg 10) labels and considering a trypsin digestion of the peptides with no cleavages on lysine or arginine before a proline. A maximum of two missed cleavages were allowed with methionine oxidation and protein N-terminal acetylation as variable modifications of proteins and carbamidomethylation as fixed modification. The maximum number of modifications allowed per peptide was set to five. Mass tolerance was set to a maximum of 7 ppm for the precursor ions and 20 ppm for the fragment ions. The minimum length of peptides to be considered for quantification was set to seven amino acids and the false discovery rate threshold set to 5%. The minimum number of peptides to be used for the identification of proteins was set to one but only proteins identified with two or more peptides were considered in further analysis. Protein quantification was calculated using both unique and razor peptides. Significant enrichment was performed using a Student $t$ test with a cutoff of $p < 0.01$.

**Absolute quantification of Ub and Ub^KEKS in cell extracts.** Exponentially growing HeLa WT cells were lysed in 8 M urea (in 10 mM HEPES). A total of 100 µg of the cell extracts were reduced in 10 mM DTT, boiled and alkylated in 7.5 mM 2-Chloroacetamide for 30 min in dark. The urea concentration in the lysate was reduced to 2 M with the addition of 50 mM $NH_4HCO_3$ and the samples were subjected to overnight trypsin digest (Trypsin Gold, MS Grade, Promega Corporation, WI, USA). Following digestion the extracted peptides were desalted using zip-tips, dried in speedvac and resuspended in 1% formic acid.

For quantification on a TimsTOF Pro mass spectrometer, samples were first loaded by HPLC (nanoElute, Bruker Daltonics) with a constant flow of 4 µl/min onto a trap column (Acclaim PepMap100 C18 column, 0.3 mm id × 5 mm, Dionex Corporation, Sunnyvale, CA). Peptides were then eluted off towards an analytical column heated to 50 °C (PepSep C18 ReproSil AQ column (75 µm × 25 cm, 1.9 µm beads size)) with a linear gradient of 5–30% of solvent B (100% acetonitrile with 0.1% formic acid) over a 30-min gradient at a constant flow (500 nl/min). Peptides were then analyzed by a TimsTOF Pro mass spectrometer (Bruker Daltonics) using a CaptiveSpray nano electrospray source at a voltage of 1.6 kV. Data were acquired using data-dependent auto-MS/MS with a 100–1700 $m/z$ mass range, with MRM enabled, $m/z$ dependent isolation window and collision energy of 42.0 eV. An inclusion list containing the $m/z$ values corresponding to the monoisotopic form of the heavy and light peptides of Ub (520.2/525.2) and Ub^KEKS (762.8/767.8) was generated. The target intensity was set to 20,000, with an intensity threshold of 2500.

For quantification on a triple-quadrupole mass spectrometer, samples were reconstituted in 20 µl $H_2O$ with 3% DMSO and 0.2% formic acid, including 5.3 ng/ml Ub internal standard (IS) peptide and 1.53 ng/ml Ub^KEKS IS peptide. Cell lysates were cleaned on Strata-X reversed phase SPE (Phenomenex) with added IS peptides (to get final concentrations 5.3 ng/ml Ub and 1.53 ng/ml Ub^KEKS). Dried samples were reconstituted in 20 µl $H_2O$ with 3% DMSO and 0.2% formic acid. Acquisition was performed with a Shimadzu LCMS-8060 (Shimadzu, Kyoto, Japan) equipped with an electrospray interface, a 100 µm ID capillary and coupled to a Nexera XR (Shimadzu, Kyoto, Japan). LabSolution v5.93 software was used to control the instrument and for data processing and acquisition. Separation was

performed on a reversed phase aeris peptide C18 100 mm × 2.1 mm (Phenomenex) over a 10-min gradient of 2–55% of solvent B (acetonitrile with 0.2% formic acid and 3% DMSO v/v). Optimized MRM parameters were used to monitor Ub and Ub$^{KEKS}$.

For quantification on an OrbiTrap mass spectrometer, peptides were loaded and separated onto a nanoHPLC system (Dionex Ultimate 3000) with a constant flow of 4 µl/min onto a trap column (Acclaim PepMap100 C18 column, 0.3 mm id × 5 mm, Dionex Corporation, Sunnyvale, CA). Peptides were then eluted off towards an analytical column heated to 40 °C (PepMap C18 nano column (75 µm × 25 cm)) with a linear gradient of 5–45% of solvent B (80% acetonitrile with 0.1% formic acid) over a 42-min gradient at a constant flow (450 nl/min). Peptides were analyzed on an OrbiTrap QExactive (Thermo Fischer Scientific) spectrometer using PRM method. Acquisition of the MS/MS spectra ($m/z$ 350–1600) was performed in the Orbitrap. An inclusion list containing the $m/z$ values corresponding to the monoisotopic form of the normal and equivalent AQUA peptides of Ub (520.2/525.2) and Ub$^{KEKS}$ (762.8/767.8) was generated (heavy Arginine (U-13C6, 15N4; mass difference: +10 Da) labeled AQUA Ub (EGIPPDQQR) and Ub$^{KEKS}$ (IQDEEGIPPDQQR) peptides (Thermo Fischer Scientific)). The collision energy was set at 28% and resolution for the MS/MS was set at 140,000 for 1,000,000 ions with maximum filling times of 250 ms with an insulation width of 0.6. Data acquisition was done using Xcalibur version 3.1.66.10.

**Absolute quantification of Ub and Ub$^{KEKS}$ on lamin A and B2.** Transient transfection was performed on wild type and Ub$^{KEKS}$ KO HeLa cells (clone 4.3) with either GFP-LMNA or GFP-LMNB2. After 48 h of transfection, cells were harvested and lysed in HS buffer for 30 min rotating at 4 °C. The lysates were then centrifuged for 10 min at 13,000 × g at 4 °C and equal amount of proteins were incubated with GFP-Trap agarose beads from (ChromoTek) overnight. Following washes elution was performed using 2xLaemli buffer supplemented with 10 mM DTT and boiled 5 min at 95 °C. Samples were then alkylated using 50 mM 2-Chloroacetamide for 30 min in dark. Following alkylation samples were loaded on 4–12% gradient SDS-PAGE gels and proteins were separated for 45 min. After separation the gel was stained with SimplyBlue SafeStain (Invitrogen) and destained in dH$_2$O overnight. Gel regions corresponding to the approximate molecular weight of monoubiquitylated GFP-LMNA or GFP-LMNB2 were excised and in-gel trypsin digestion was performed. Following digestion the extracted peptides were desalted using zip-tips, dried in speedvac and resuspended in 1% formic acid. Heavy Arginine (U-13C6, 15N4; mass difference: +10 Da) labeled AQUA Ub (EGIPPDQQR) and Ub$^{KEKS}$ (IQDEEGIPPDQQR) peptides (Thermo Fischer Scientific) were spiked into the samples at a final concentration of 1.66 fmol/µl. Peptides were loaded and separated onto a nanoHPLC system (Dionex Ultimate 3000) with a constant flow of 4 µl/min onto a trap column (Acclaim PepMap100 C18 column, 0.3 mm id × 5 mm, Dionex Corporation, Sunnyvale, CA). Peptides were then eluted off towards an analytical column heated to 40 °C (PepMap C18 nano column (75 µm × 25 cm)) with a linear gradient of 5–45% of solvent B (80% acetonitrile with 0.1% formic acid) over a 42-min gradient at a constant flow (450 nl/min). Peptides were analyzed on an OrbiTrap QExactive (Thermo Fischer Scientific) spectrometer using PRM method. Acquisition of the MS/MS spectra ($m/z$ 350–1600) was performed in the Orbitrap. An inclusion list containing the $m/z$ values corresponding to the monoisotopic form of the heavy and light peptides of Ub (520.2/525.2) and Ub$^{KEKS}$ (762.8/767.8) was generated. The collision energy was set at 28% and resolution for the MS/MS was set at 140,000 for 1,000,000 ions with maximum filling times of 250 ms with an insulation width of 0.6. Data acquisition was done using Xcalibur version 3.1.66.10.

Identification and quantification of Ub and Ub$^{KEKS}$ peptides was performed on Skyline software (19.1.0.193)[69]. For quantification only the most intense fragment ion (y6) was used for both peptides. The amount of Ub and Ub$^{KEKS}$ proteins were calculated using the light to heavy peptide ratio.

**Reporting summary.** Further information on research design is available in the Nature Research Reporting Summary linked to this article.

## Data availability

The mass spectrometry proteomics data were deposited to the ProteomeXchange Consortium via the PRIDE partner repository with the dataset identifier PXD014318. The source data underlying Figs. 2c–e, 3a, 4a, b, d–g, 5c, and Supplementary Fig. 8 are provided as a Source Data file. All other data are available from the corresponding authors on reasonable request.

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

## Acknowledgements

We thank the proteomics facility of the Université de Sherbrooke for assistance with mass spectrometry. M.-L.D. is a recipient of an Alexander Graham Bell NSERC studentship. F.-M.B is a recipient of a FRQS Junior 2 scholarship. X.R. is a recipient of a Canada Research Chair in Functional Proteomics and Discovery of Novel Proteins. Funding was provided from the Canadian Institutes for Health Research, grant number #398925 to P.L., X.R., and F.-M.B. Authors P.L., X.R., M.S.S., and F.M.B. are members of the FRQS-funded "Centre de Recherche du CHUS".

## Author contributions

S.S., M.S.S., F.-M.B., and X.R. made the initial discovery of UBBP4 as a potentially expressed protein. M.-L.D., F.-M.B., and X.R. designed the experimental approaches. M.-L.D. and F.-M.B. prepared Fig. 1. F.-M.B. prepared Fig. 2a, b and Supplementary Figs. 1–3, 5. M.-L.D. performed the experiments in Figs. 2c, d, 3a, b, and 4b–g and Supplementary Fig. 6. A.M. performed all the absolute quantification, including Fig. 5 and Supplementary Fig. 9, as well as the rescue in Supplementary Fig. 8. M.C.B. performed Fig. 3a. M.A.B. and J.-F.J. performed the experiment in Fig. 2e. M.A.B. performed the statistical analysis of Fig. 3c and Supplementary Data 1. M.B. generated the CRISPR KO cells. J.F. performed the experiment in Fig. 4a. A.T. performed the experiment in Supplementary Fig. 4. S.S. prepared Supplementary Fig. 7. D.L. prepared some of the plasmids used in this study and setup the CFSE cell growth assays. F.-M.B. and M.-L.D. wrote the main paper text.

## Competing interests

The authors declare no competing interests.

## Additional information

**Peer review information** *Nature Communications* thanks the anonymous reviewers for their contribution to the pee review of this work. Peer reviewer reports are available.

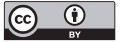

