## [Peer Review File · Nature Communications]

Reviewers' comments:

Reviewer #1 (Remarks to the Author):

In this manuscript the author(s) report on the expression of a ubiquitin pseudogene, UBBP4, as well as properties and potential involvement of the resulting protein. The repertoire of ubiquitin genes is complex, as it encompasses several pseudogenes, and the authors provide evidence from published mass spectrometry data sets that some of these pseudogenes actually result in protein. Given the complexity of the ubiquitin code even with "wildtype" ubiquitin, the realisation that sequence variants with potentially altered properties are expressed in cells is really exciting, as this would add another layer of complexity and possibly regulation to the ubiquitin world.

The authors focus their analysis on a ubiquitin variant (Ub-KEKS) that is encompassed within the UBBP4 pseudogene and differs from ubiquitin in 4 amino acids. Using an overexpressed HA-tagged version they show conjugation to cellular proteins and differences in the localisation as well as relationship to the proteasomal system. In a SILAC-MS approach they identify proteins preferentially modified by the variant, among them the lamins, which is consistent with the localisation data. They then analyse two independent knockout clones of Ub-KEKS and show that these cells have a growth defect as well as slightly enlarged nucleoli.

Overall, the manuscript is intriguing and potentially very exciting, but it is also rather preliminary. I can understand the urge to publish these findings quickly, possibly due to competition in the field. At the same time, a validation of the results by additional control experiments would be necessary to support a physiological relevance of the observed phenomena. My detailed comments are as follows:

1. The only two pieces of evidence for a physiological relevance of Ub-KEKS are the growth defect and the (slight) enlargement of the nucleoli in the knockout clones. In order to rule out a spurious effect, I would consider it necessary to provide some kind of complementation data, i.e. to rescue the phenotypes by means of re-expressing the variant in the knockout cells. I realise that this might be difficult if protein levels matter, but as it stands now, the evidence for a functional significance is thin, because the authors cannot rule out that the observed defect is based on an effect of the RNA. The author(s) could try to express either a functional or a delta-GG version of Ub-KEKS and try to observe at least a rescue of the growth defect. Another way to rule out at least the RNA effect would be to do a more subtle CRISPR/Cas manipulation of the sequence to introduce a premature stop codon, preferentially late in the gene, e.g. by generating something like the delta-GG version, instead of removing such a large piece of sequence.

2. It would be important to provide some measure of the balance between WT and variant ubiquitin, both in terms of total levels and relating to the preferred substrates, under non-overexpressing conditions. In other words, how abundant is the variant protein in cells overall when compared to ubiquitin, and what is the ratio on the lamins or other relevant substrates? In principle, this should be doable by mass spectrometry, by searching for the relevant peptides. This kind of information would be extremely helpful in assessing the physiological relevance of the modification. In its absence, a spurious modification at an extremely low level that may affect individual molecules to some degree, but has no bearing on the overall substrate pool, cannot really be ruled out.

3. There are of course all sorts of interesting biochemical questions about Ub-KEKS, i.e. does it form chains? Which E2s/E3s are involved? As far as I know, E3s for lamins have been identified - do they exhibit a preference for Ub-KEKS? Are lamins less ubiquitylated in the knockout cells? And so on. Answering all these questions is clearly beyond the scope of the current manuscript, and I am sure the author(s) are considering these types of experiments, but even some attempts at looking for phenotypes associated with the targets or the conjugation factors that might resemble those of Ub-KEKS knockout would be helpful to substantiate the conclusions made in this manuscript.

Reviewer #2 (Remarks to the Author):

The manuscript entitled “UBB Pseudogene 4 encodes novel ubiquitin variants” describes an interesting exploration of unexpected protein variants of ubiquitin emanating from the many pseudogenes scattered across the genome. In this work, the authors provide convincing evidence for the existence of translated polypeptides of UBBP4. The paper goes on to show that two exogenously expressed UBBP4 variants can undergo conjugation to target proteins, further expanding the diversity of covalent ubiquitin-like modifications within the human proteome.

What is missing from this study is some demonstration that endogenously expressed UBBP4 variants such as B1 or A2 can be conjugated to protein substrates. Given the identification of lamins as key targets of variant B1, one way this could be done would be to pull down tagged lamin in cells endogenously expressing UBBP4-B1 and detect one or more variant specific peptides.

This would be most convincing if done quantitatively alongside peptides that distinguish between WT and variant Ub to provide an early assessment of the relative abundance of these two related modifications. This is important to support the major claim that a “subset of proteins reported as Ub targets could rather be modified by Ub variants”.

In Figure 1E, why does the UBBP4-B1 not localize appear to localize to the nucleus or the nuclear lamina?

How many replicates were performed for the data in figure 3C-D?

The data in Fig 3D was easy to understand, but 3C was much less so. Why did the authors choose to plot Z-score scatter plots and not the more natural log₂ ratios? The importance and reliability of the 90th percentile cutoffs was unclear. Limiting the axes to show only the areas of interest seems like a questionable decision. Overall, the authors should consider a more informative way of displaying these data.

Do the authors see evidence for the intact Cys on the C-terminus of UBBP4-B1? Does this participate in conjugation of this variant onto protein substrates? Was this C-terminal Cys included in the HA-tagged construct that was expressed?

On p3 the text extrapolates from the MG132 data that “proteins modified by UBBP4-B1 are not targeted for proteasomal degradation”. This is the logical interpretation, but that has not been formally proven by these data.

The authors should compare the sequences of UBBP4 variants to the sequences emerging from the Sidhu lab in Toronto on directed evolution and combinatorial mutagenesis of Ub to see if any of the pseudogenes mirror sequences emerging from the engineered mutants.

Response to Referees

Reviewer #1 (Remarks to the Author):

In this manuscript the author(s) report on the expression of a ubiquitin pseudogene, UBBP4, as well as properties and potential involvement of the resulting protein. The repertoire of ubiquitin genes is complex, as it encompasses several pseudogenes, and the authors provide evidence from published mass spectrometry data sets that some of these pseudogenes actually result in protein. Given the complexity of the ubiquitin code even with "wildtype" ubiquitin, the realisation that sequence variants with potentially altered properties are expressed in cells is really exciting, as this would add another layer of complexity and possibly regulation to the ubiquitin world.

The authors focus their analysis on a ubiquitin variant (Ub-KEKS) that is encompassed within the UBBP4 pseudogene and differs from ubiquitin in 4 amino acids. Using an overexpressed HA-tagged version they show conjugation to cellular proteins and differences in the localisation as well as relationship to the proteasomal system. In a SILAC-MS approach they identify proteins preferentially modified by the variant, among them the lamins, which is consistent with the localisation data. They then analyse two independent knockout clones of Ub-KEKS and show that these cells have a growth defect as well as slightly enlarged nucleoli.

Overall, the manuscript is intriguing and potentially very exciting, but it is also rather preliminary. I can understand the urge to publish these findings quickly, possibly due to competition in the field. At the same time, a validation of the results by additional control experiments would be necessary to support a physiological relevance of the observed phenomena.

We agree with the reviewer. The goal of this manuscript is to demonstrate beyond any reasonable doubts that Ub^{KEKS} does exist, and can be attached to other proteins as a prototype of Ub variants encoded by 'pseudogenes'. We believe that we made a strong case for this small protein to be expressed, and to be functional in attaching to other specific proteins that are different from Ub, and without targeting them for degradation. With the help of the comments made by the reviewers, we believe that the current manuscript now includes additional information particularly on the quantification of Ub^{KEKS}, and that this strengthened version of the manuscript will lead to high impact findings that will change significantly the way people are studying ubiquitylation.

My detailed comments are as follows:

1. The only two pieces of evidence for a physiological relevance of Ub-KEKS are the growth defect and the (slight) enlargement of the nucleoli in the knockout clones. In order to rule out a spurious effect, I would consider it necessary to provide some kind of complementation data, i.e. to rescue the phenotypes by means of re-expressing the variant in the knockout cells. I realise that this might be difficult if protein levels matter,

but as it stands now, the evidence for a functional significance is thin, because the authors cannot rule out that the observed defect is based on an effect of the RNA. The author(s) could try to express either a functional or a delta-GG version of Ub-KEKS and try to observe at least a rescue of the growth defect. Another way to rule out at least the RNA effect would be to do a more subtle CRISPR/Cas manipulation of the sequence to introduce a premature stop codon, preferentially late in the gene, e.g. by generating something like the delta-GG version, instead of removing such a large piece of sequence.

As suggested, we performed a rescue experiment by transfecting HA-Ub^{KEKS} in the two different Ub^{KEKS} KO cell lines. This is now shown as supplementary Figure 8. 72 hours post-transfection, cells positive for HA no longer had lamins within the nucleolus, and the size of the nucleolus was also back to normal. An example of a field is shown, with higher magnification of the nucleolus shown with arrows.

2. It would be important to provide some measure of the balance between WT and variant ubiquitin, both in terms of total levels and relating to the preferred substrates, under non-overexpressing conditions. In other words, how abundant is the variant protein in cells overall when compared to ubiquitin, and what is the ratio on the lamins or other relevant substrates? In principle, this should be doable by mass spectrometry, by searching for the relevant peptides. This kind of information would be extremely helpful in assessing the physiological relevance of the modification. In its absence, a spurious modification at an extremely low level that may affect individual molecules to some degree, but has no bearing on the overall substrate pool, cannot really be ruled out.

We performed absolute quantification of Ub and Ub^{KEKS} using AQUA peptides spiked either in whole cell lysates, as well as Lamin A and B2 immunoprecipitations (Fig. 5, Supplementary Fig. 9, and Supplementary Tables 4 and 5). We measured Ub at 142 fmol/ug, consistent with previous publications using a similar approach, and measured Ub^{KEKS} at 0.23 fmol/ug. While the ratio appears to be rather large, the amount of Ub^{KEKS} in cells is close to SUMO1 and can still be considered an above-average abundant protein. This quantification was performed in duplicate using 3 different instruments. Two that we have in our facility (OrbiTrap QExactive and TimsTOF Pro), and one as a service from an outside company (PhenoSwitch Bioscience), using a Triple-Quadrupole (Shimadzu LCMS-8060). While the TimsTOF is obviously not well suited for these type of quantification and provided noisier quantification, all three instruments gave similar quantification, and we have thus high confidence in the measurements we present. It remains to be seen whether Ub^{KEKS} is competing with Ub for targets, whether it has specific targets, or whether it can form chains with Ub and/or act as a chain terminator. However, the amount of Ub^{KEKS} at Lamins is much higher compared to the cells, indicating a specificity for modification of Lamins.

3. There are of course all sorts of interesting biochemical questions about Ub-KEKS, i.e. does it form chains? Which E2s/E3s are involved? As far as I know, E3s for lamins have been identified - do they exhibit a preference for Ub-KEKS? Are lamins less ubiquitylated in the knockout cells? And so on. Answering all these questions is clearly beyond the scope of the current manuscript, and I am sure the author(s) are considering

these types of experiments, but even some attempts at looking for phenotypes associated with the targets or the conjugation factors that might resemble those of Ub-KEKS knockout would be helpful to substantiate the conclusions made in this manuscript.

Indeed, because the article was originally submitted to Nature and transferred as is to Nature Communications for review, it did not include a proper introduction and discussion, which certainly raises many questions as a result of the identification of new ubiquitin variants. We did identify some K49 modification on Ub^{KEKS} itself, and this information is in the peptide files in the PRIDE repository. This obviously remain to be confirmed endogenously, hence why we did not want to provide this evidence because it was observed following overexpression of HA-Ub^{KEKS}, but opens up very interesting possibilities of new ubiquitin chains. Also, the identification of known Ub targets such as Lamins, PCNA and histones suggest that perhaps some of the known E2 or E3 ligases could specifically select for Ub^{KEKS}, moreover, our AP-MS identified several E2 and E3 that were consistently found in all repeats. These will have to be pursued and tested in vitro to at least demonstrate specificity for Ub and Ub^{KEKS}. These information have been added to the discussion, which now properly addresses these questions.

Reviewer #2 (Remarks to the Author):

The manuscript entitled “UBB Pseudogene 4 encodes novel ubiquitin variants” describes an interesting exploration of unexpected protein variants of ubiquitin emanating from the many pseudogenes scattered across the genome. In this work, the authors provide convincing evidence for the existence of translated polypeptides of UBBP4. The paper goes on to show that two exogenously expressed UBBP4 variants can undergo conjugation to target proteins, further expanding the diversity of covalent ubiquitin-like modifications within the human proteome.

What is missing from this study is some demonstration that endogenously expressed UBBP4 variants such as B1 or A2 can be conjugated to protein substrates. Given the identification of lamins as key targets of variant B1, one way this could be done would be to pull down tagged lamin in cells endogenously expressing UBBP4-B1 and detect one or more variant specific peptides.

This would be most convincing if done quantitatively alongside peptides that distinguish between WT and variant Ub to provide an early assessment of the relative abundance of these two related modifications. This is important to support the major claim that a “subset of proteins reported as Ub targets could rather be modified by Ub variants”.

We totally agree with the proposed quantification, which was also suggested by reviewer #1. We performed absolute quantification of Ub and Ub^{KEKS} in whole cell lysates, as well as on Lamin pull down (see answer to reviewer #1 for details). This

provides further evidence that endogenous Ub^{KEKS} can be conjugated to protein substrates, and confirms the specificity for targets such as Lamins.

In Figure 1E, why does the UBBP4-B1 not localize appear to localize to the nucleus or the nuclear lamina?

We assume the reviewer meant Fig. 2E. When comparing Ub to Ub^{KEKS} (or Ubbp4^{B1}), Ub^{KEKS} appears to be more prominent in the nucleus, but also shows a distinctive signal at the nuclear periphery, consistent with the nuclear lamina.

How many replicates were performed for the data in figure 3C-D?

These experiments were performed in biological triplicates. This information has been added to the methods section, as well as in the figure legend.

The data in Fig 3D was easy to understand, but 3C was much less so. Why did the authors choose to plot Z-score scatter plots and not the more natural log₂ ratios? The importance and reliability of the 90th percentile cutoffs was unclear. Limiting the axes to show only the areas of interest seems like a questionable decision. Overall, the authors should consider a more informative way of displaying these data.

We thought using a statistical cutoff with a z-score would provide a better rationale for the proteins we display in figure 3, as opposed to choosing an arbitrary cutoff of 2-fold for example. This type of statistical analysis is also now required by Nature for compliance with editorial policies. As an example, the cutoff we chose correspond to an average enrichment of 5-fold for proteins modified by Ub^{KEKS} over the control IP (n=3), and thus provides a very confident shortlist of proteins to display in Figure 3. However, the Supplementary Table 2 has all the data available with all the proteins sorted by their enrichment ratio, and thus provide the information if someone is interested in an extended list of possible targets. The same proteins are displayed as a dot plot in Figure 3d, which we agree is a lot more convivial, but we thought it was important to provide a statistical analysis of the protein list.

Do the authors see evidence for the intact Cys on the C-terminus of UBBP4-B1? Does this participate in conjugation of this variant onto protein substrates? Was this C-terminal Cys included in the HA-tagged construct that was expressed?

Interestingly, UBB also has a cysteine at its proximal ubiquitin, and the C-terminal extension of UBB can be removed by DUBs such as USP5. We did initially test both a plasmid encoding HA-tagged Ub^{KEKS} with and without the C-terminal cysteine, and observed no differences in the pattern, as well as the level of expression. In order to compare ubiquitin with Ub^{KEKS} without the interference of this cysteine, which is likely removed, we decided to not include the C-terminal cysteine in the HA-tagged construct used in the article. This information has been added in the method section. Moreover, proteins modified by Ub^{KEKS} identified by mass spectrometry had a typical GlyGly attached to Lysines, and as such does not provide any evidence for other type of modification. The immunoblot presented in the article were performed by denaturing SDS-PAGE in the presence of DTT, and would not display di-sulfide bonds.

On p3 the text extrapolates from the MG132 data that “proteins modified by UBBP4-B1 are not targeted for proteasomal degradation”. This is the logical interpretation, but that has not been formally proven by these data.

We agree that our data do not formally demonstrate this point. This has been toned down in the abstract, and on p.3 to say that our data support the observation that proteins are not targeted for proteasomal degradation, instead of being conclusive.

The authors should compare the sequences of UBBP4 variants to the sequences emerging from the Sidhu lab in Toronto on directed evolution and combinatorial mutagenesis of Ub to see if any of the pseudogenes mirror sequences emerging from the engineered mutants.

This is a very interesting suggestion. The residues which are different in Ub^{KEKS} (2, 33, 49 and 60) do not seem to be putative inhibitors for RING or HECHT E3 ligases as identified in studies from Sidhu’s lab, but both residues 2 and 49 appear to be amongst the top hits for ubiquitin-specific proteases, which could result in modification by Ub^{KEKS} to be partial resistant to DUBs. It will be crucial in follow-up studies to mutate the 4 residues that are different in Ub^{KEKS} back to Ub to determine which residue(s) are necessary for the difference in specificity of targets. This has been added to the discussion.

Reviewers' comments:

Reviewer #1 (Remarks to the Author):

The authors have substantiated their findings by a set of new data and thereby addressed the key concerns that were raised before. I am now happy with the manuscript as it stands.

Minor point: line 227: HECT (not HECHT)

Reviewer #2 (Remarks to the Author):

In the revised manuscript from Dubois et.al., the authors have provided additional experimental data to solidify their claim that the UBBP4 pseudogene is expressed at the protein level and conjugated to substrates including lamin A/B. By and large, the concerns expressed as part of my initial review have been addressed either by these data or the text revisions inasmuch that I generally believe the authors have probably discovered something new and interesting.

With that said, there are aspects of the new data that do not necessarily make sense. Additionally, there are elements of the original data that remain unclear and do not support the conclusions as strongly they should.

1. Figure 4 has shows phenotypic data from two CRISPR ko clones. The phenotypes described include delayed cell growth, lamin subcellular distribution, and nucleolar size. The latter two are subtle imaging phenotypes, where the Lamin staining data in 4E were not quantified to support the claim that Lamin A accumulated. Moreover, all three observations would be solidified if rescue data were available showing phenotypic reversal upon re-expression. Given the inherent clonal variability, such an experiment seems prudent. The data in Suppl Fig 8 are described as 72 hr re-expression rescue experiments but are not sufficiently informative to address this request.

2. Quantitative MS experiments were done using AQUA/SIL peptides bearing ¹³C₆-¹⁵N₄-Arg labelled residues to measure the levels of Ub and Ub-KEKS at the key locus surrounding Lys-33 of ubiquitin. An expectation with ¹³C and ¹⁵N labeled peptides is that the heavy and light analytes should co-elute perfectly. The data in Fig 5B shows the unlabeled Ub-KEKS peptide with the sequence IQDEEGIPPDQQR eluting as much 30 sec earlier and with a much narrower peak width than the isotope labeled standard. Moreover, this RT differential is inconsistent with between the HeLa and HeLa 4.3 cells.

This means that one or the other is not what the authors think it is. What do the MS2 spectra look like side by side at different points across the elution peak...for example at the a) apex of the red trace, at the b) apex of the blue trace, and at the c) tail of the blue trace. How do the authors explain this RT difference? What do the precursor ion isotopic envelopes look like at each of these timepoints...are the authors comparing equivalent mono-isotopic peaks, or is there possibly a 1 Da register error with one or the other chromatographic feature?

The contributing factor here may be that the y6 proline peak is used as the single fragment ion for multiple features. Having a b-series ion within the PRM analysis could help sort out specific/intended signals from those emerging from related by non-identical features.

The bottom line is that something is not as it seems.

3. Again considering Fig 5B, the HeLa 4.3 cell line is a CRISPR ko line that lacks Ub-KEKS. Why does any signal remain here? Does sequencing data confirm the presence of some residual Ub-KEKS mRNA? Is there evidence to show that the genomic locus of Ub-KEKS remains intact in some residual population of cells? To see the Ub-KEKS protein level go down by only 2-3 fold in a CRISPR ko cell line is concerning and further amplifies concerns that something is amiss.

4. Assuming points 2-3 above can be explained in a satisfactory manner, more information should be provided to describe how the peak areas were determined for these unusually shaped peaks. It is well established that the peptide sequence of Ub (and presumably Ub-KEKS by extension) containing the EGIPPDQQR sequence displays peak splitting, owing to on column proline isomerization. This makes peak area boundaries critical variables that much be defined explicitly in quantitative MS analysis of these features. The conclusions of the quantitative data could vary dramatically depending on how this process was done.

5. The Discussion text on p9 has a very speculative section postulating a differential role for S5A between Ub and Ub-KEKS. Since there are multiple Ub-binding domain receptors, and no evidence that S5A is involved here, this section felt like an overreach even for a discussion section.

Response to referees letter

In the revised manuscript from Dubois et.al., the authors have provided additional experimental data to solidify their claim that the UBBP4 pseudogene is expressed at the protein level and conjugated to substrates including lamin A/B. By and large, the concerns expressed as part of my initial review have been addressed either by these data or the text revisions inasmuch that I generally believe the authors have probably discovered something new and interesting.

With that said, there are aspects of the new data that do not necessarily make sense. Additionally, there are elements of the original data that remain unclear and do not support the conclusions as strongly they should.

1. Figure 4 has shows phenotypic data from two CRISPR ko clones. The phenotypes described include delayed cell growth, lamin subcellular distribution, and nucleolar size. The latter two are subtle imaging phenotypes, where the Lamin staining data in 4E were not quantified to support the claim that Lamin A accumulated. Moreover, all three observations would be solidified if rescue data were available showing phenotypic reversal upon re-expression. Given the inherent clonal variability, such an experiment seems prudent. The data in Suppl Fig 8 are described as 72 hr re-expression rescue experiments but are not sufficiently informative to address this request.

Indeed, we did observed a rescue 48 and 72 hours after transfection, but not 24 hours post-transfection. Our guess is that the difference observed in lamins localization depends on going through mitosis for the nucleus to break down and reform, but as this is highly speculative, we decided not to discuss this further.

2. Quantitative MS experiments were done using AQUA/SIL peptides bearing ¹³C6-¹⁵N4-Arg labelled residues to measure the levels of Ub and Ub-KEKS at the key locus surrounding Lys-33 of ubiquitin. An expectation with ¹³C and ¹⁵N labeled peptides is that the heavy and light analytes should co-elute perfectly. The data in Fig 5B shows the unlabeled Ub-KEKS peptide with the sequence IQDEEGIPDQQR eluting as much 30 sec earlier and with a much narrower peak width than the isotope labeled standard. Moreover, this RT differential is inconsistent with between the HeLa and HeLa 4.3 cells.

This means that one or the other is not what the authors think it is. What do the MS2 spectra look like side by side at different points across the elution peak...for example at the a) apex of the red trace, at the b) apex of the blue trace, and at the c)tail of the blue trace. How do the authors explain this RT difference? What do the precursor ion isotopic envelopes look like at each of these timepoints...are the authors comparing equivalent mono-isotopic peaks, or is there possibly a 1 Da register error with one or the other chromatographic feature?

The contributing factor here may be that the y6 proline peak is used as the single fragment ion for multiple features. Having a b-series ion within the PRM analysis could help sort out specific/intended signals from those emerging from related by non-identical features.

The bottom line is that something is not as it seems.

We agree that the MS quantification is an important aspect of the manuscript, and that some details on the development of the quantification approach were not sufficiently presented. Because of that, the choice of the peptide for quantification, the ion selection, and the presence of an early-eluting peak raises some concerns regarding the validity of the quantification.

We analysed the product ions of the synthetic peptides to identify which ion showed the highest intensity (new supplementary Figure 9a). While the y6++, y5 and y6 ion intensities were relatively high for Ub, only the y6 seemed suitable for quantification for UbKEKS. To be consistent, we chose one ion for each of Ub and UbKEKS for quantification. Because the AQUA peptide bearing the 13C6-15N4 residue has the labelled arginine at its C-terminal, it is not possible to quantify using b-series ions as suggested.

Regarding the signal that is not co-eluting: This peak is not observed when the heavy peptide is analysed alone, and is only present when a cell extract is added (new supplementary Figure 9b). This was confirmed to not be related to UbKEKS, as the signal of that early eluting peak did not increase when analysing a whole cell extracts from cells overexpressing UbKEKS (new supplementary Figure 9b), while the signal for UbKEKS co-eluting with the heavy peptide did increase dramatically. This confirmed that the early eluting peak is originating from the cell extracts, but is not UbKEKS.

3. Again considering Fig 5B, the HeLa 4.3 cell line is a CRISPR ko line that lacks Ub-KEKS. Why does any signal remain here? Does sequencing data confirm the presence of some residual Ub-KEKS mRNA? Is there evidence to show that the genomic locus of Ub-KEKS remains intact in some residual population of cells? To see the Ub-KEKS protein level go down by only 2-3 fold in a CRISPR ko cell line is concerning and further amplifies concerns that something is amiss.

Because we quantified Ub^{KEKS} on shorter gradients in order to increase the peak intensity of Ub^{KEKS} by reducing the peak width, we did have a slight overlap in signal from that early eluting contaminant, which does provide some signal when performing the quantification. As demonstrated on the LMNA pulldown experiments comparing the WT and KO cells, we can see that the peak corresponding to Ub^{KEKS} in the WT cells is perfectly co-eluting with the AQUA peptide (black arrow), and this peak is not observed in the KO cells (new supplementary Figure 9c). However, the quantification does not measure a complete KO of Ub^{KEKS}, because we did have some signal trailing from that early-eluting peak (see the red trail left in the HeLa 4.3). We can thus conclude that cells that are KO for Ub^{KEKS} are not likely to have remnants of Ub^{KEKS}, which was also confirmed by genomic sequencing.

4. Assuming points 2-3 above can be explained in a satisfactory manner, more information should be provided to describe how the peak areas were determined for these unusually shaped peaks. It is well established that the peptide sequence of Ub (and presumably Ub-KEKs by extension) containing the EGIPPDQQR sequence displays peak splitting, owing to on column proline isomerization. This makes peak area boundaries critical variables that much be defined explicitly in quantitative MS analysis of these features. The conclusions of the quantitative data could vary dramatically depending on how this process was done.

See answer above for peak selection and gradient length. Because the early eluting peak is not increasing following overexpression of UbKEKS, we do not believe it is the result of peak splitting.

5. The Discussion text on p9 has a very speculative section postulating a differential role for S5A between Ub and Ub-KEKS. Since there are multiple Ub-binding domain receptors, and no evidence that S5A is involved here, this section felt like an overreach even for a discussion section.

We agree of the highly speculative nature of this part of the discussion, and we removed the reference to S5A recognition.

REVIEWERS' COMMENTS:

Reviewer #2 (Remarks to the Author):

I commend the authors for their thoughtful handling of the follow up questions. Their responses demonstrate careful consideration of the complex issues surrounding quantitation of the Ub and Ub-KEKS specific features. Updates to the text and figures provide clarity to the reader about what signal the quantitative data derive from. I feel that the data as a whole supports their conclusions and merits publication.

As advice for future work, I would encourage the authors to acquire a different AQUA peptide for Ub-KEKS that carries multiple isotopic labels or positions the label in a way that alters the reporter ions for unambiguous peak identification/quantification.